# The Heart–Brain Axis in the Artificial Intelligence Era: Integrating Old and New Insights Towards New Targeting and Innovative Neuro- and Cardio-Therapeutics

**DOI:** 10.3390/ijms26178217

**Published:** 2025-08-24

**Authors:** Andreas Palantzas, Maria Anagnostouli

**Affiliations:** 1Athens School of Medicine, National and Kapodistrian University of Athens, NKUA, 106 79 Athens, Greece; andreasplntzs@gmail.com; 2Multiple Sclerosis and Demyelinating Diseases Unit, Center of Expertise for Rare Demyelinating and Autoinflammatory CNS Diseases, 1st Department of Neurology, School of Medicine, National and Kapodistrian University of Athens, NKUA, Aeginition University Hospital, 115 28 Athens, Greece

**Keywords:** heart, brain, autonomic nervous system, neuroimmune cardiovascular interface, neuro endocrino immunological system, multiple sclerosis, wearables, artificial intelligence, antistress techniques

## Abstract

The heart–brain axis (HBA) is a dynamic system of reciprocal communication between the cardiovascular and central nervous system, incorporating neural, immunologic, molecular and hormonal pathways. The central autonomic network is described as a key regulator of cardiovascular activity and autonomic dysfunction as an important mechanism underlying various neurologic and cardiac disorders. Heart rate variability (HRV) is identified as the key biomarker of the axis reflecting autonomic nervous system balance. Increased understanding of its molecular mechanisms has led to the proposal of new therapeutic strategies focused on modulating heart–brain communication including β-blockers, vagus nerve stimulation, neurotrophin modulation, and nanoparticle-based approaches. The integration of wearables and artificial intelligence (AI) has allowed for real-time monitoring and innovative diagnostic and prognostic applications. The present narrative review summarizes current knowledge on the mechanisms comprising the heart–brain axis, their implication in neurologic and cardiac disorders, and their potential for developing novel therapies. It also highlights how advancements in wearable technology and AI systems are being integrated into clinical practice and transforming the landscape.

## 1. Introduction

The heart–brain axis (HBA) is a dynamic bi-directional communication network that incorporates neural, immunologic, molecular, and hormonal pathways responsible for regulating the functions of both the cardiovascular and central nervous system (CNS) [1,2,3,4,5]. The relationship between the heart and the brain has been studied since ancient times. Across history, many prominent figures have attempted to interpret it, from Alcmeon and Plato to Darwin and James-Lang. Today, it has been established that communication between these two organs is primarily mediated by the autonomic nervous system (ANS), specifically by a set of cortical and subcortical structures termed the central autonomic network (CAN) that coordinates afferent and efferent signals [4,5,6]. Heart rate variability (HRV) has emerged as the key biomarker of the axis reflecting autonomic balance and offering diagnostic and prognostic insights into both neurologic and cardiovascular disorders [5,7].

In a clinical context, dysregulation of the HBA, most frequently manifesting as sympathetic hyperactivity or autonomic failure, has been identified as a key underlying mechanism in many disorders including heart failure (HF), stroke, hypertension (HTN), multiple sclerosis (MS), takotsubo cardiomyopathy (TC), and depression [5,8,9,10,11,12,13,14].

Recent advances have shed light onto the molecular mechanisms underlying the complex HBA pathways, highlighting the importance of inflammatory cytokines (e.g., IL-6 and TNF-α), neurotrophins, circulating RNAs, and even microbial metabolites and gut barrier integrity [9,15,16,17,18,19,20,21,22]. This has paved the way for the development of novel therapeutic approaches that strategically target HBA pathways including the β-blockade, vagus nerve stimulation, neurotrophin modulation, anti-inflammatory agents, and nanoparticle-based delivery systems [8,9,10,16,18,23,24].

In addition, the rise in wearable technology and artificial intelligence (AI) offers transformative potential for innovative diagnostic, prognostic, and monitoring applications. Wearable devices measuring HBA biomarkers like HRV, electrocardiographic (ECG) changes, troponin, and brain natriuretic peptide (BNP) are increasingly coupled with AI systems, enabling early detection, real-time monitoring, and personalized interventions across HBA-related disorders [8,10].

The present review aims to summarize the current knowledge on the history of the HBA initiative insights, the mechanisms and molecular pathways that comprise it, and their role in neurologic and cardiovascular disorders. It presents how these insights can be leveraged and translated into novel therapies emphasizing the role of nanoparticles and highlights how the integration of wearables and AI systems are redefining the standard of care in diagnosis, treatment, and precision medicine.

## 2. The Heart–Brain Axis Throughout History

The idea that a connection exists between the brain and the heart can be traced back to ancient civilizations. The ancient Egyptians along with the greatest Greek philosophers like Plato and Aristotle were among the first to talk about this relationship. Hippocrates was the first to describe epileptic seizures, and Alcmaeon of Croton, a pre-Socratic philosopher and student of Pythagoras, considered the brain to be the center of sensation, movement, and thought, and additionally, he was the first to emphasize the brain as the center of mental functions, in contrast to other theories that emphasized the heart. Later, Galen was the first to describe the sympathetic chain and the first to demonstrate that peripheral nerves were mostly mixed, while in parallel there were nerves that were only motor or only sensory, among other brain diseases’ descriptions [1].

Then, Bodechtel and Aschnebrenner (1938) and Byer & Co. (1947) were among the first to document electrocardiographic changes in stroke patients, marking some of the earliest clinical observations that linked cerebrovascular events to cardiac function [8].

In 1963, Melvile et al. uncovered the role of the hypothalamus in regulating cardiac autonomic activity, and during the 1980s and 1990s, contributions from Hachinski, Oppenheimer, and Cechetto regarding the insula led to the rise of the lateralization hypothesis, which associated the right insula with sympathetic and the left with parasympathetic activity [3,8,25]. The role of areas like the insula, anterior cingulate cortex, and amygdala were eventually established with the use of neuroimaging techniques like PET and fMRI, which also confirmed the lateralization hypothesis [8,25].

During the same period, the concept of the central autonomic network (CAN) was introduced, and by 1985, Nelson became the first to use the term neurocardiology to describe the interdisciplinary study of heart–brain interactions.

Today, the concepts of interoception and cardioception have become central in understanding heart–brain signaling [4,5,26], and the heart–brain axis is now viewed as a dynamic system that involves reciprocal communication via both neural and humoral pathways, with HRV as the key biomarker reflecting the integrity of this communication [2,4,5,7].

## 3. Description of the Heart–Brain Axis

### 3.1. Autonomic Nervous System

The autonomic nervous system (ANS) is the main mediator of central control over the heart. Through the CAN—a set of cortical and subcortical structures—the ANS sends efferent sympathetic and parasympathetic outputs to the heart and receives afferent feedback via spinal and vagal pathways [6]. These structures are as follows [6,27,28,29,30,31,32] (Figure 1):

**Cerebral cortex**: insular cortex, medial prefrontal cortex (MPFC), anterior cingulate cortex, orbitofrontal cortex, posterior cingulate cortex

**Limbic system**: amygdala, bed nucleus of the stria terminalis (BNST)


**Diencephalon and brainstem**


Forebrain: thalamus, hypothalamus (paraventricular nucleus-PVN, dorsomedial hypothalamus-DMH)

Midbrain: periaquadactal gray (PAG)

Pons: parabrachial nucleus, Kölliker–Fuse, A5 area

Medulla: nucleus tractus solitarius (NTS), rostral ventrolateral medulla (RVLM), rostral ventromedial medulla (RVMM), nucleus ambiguus (NAmb), dorsal motor nucleus of the vagus (DMNX)

The sympathetic pathway starts from preganglionic neurons (S1N) in the upper thoracic spinal cord (T1–T5), which then synapse in the cervical and thoracic ganglia (S2N) [33,34]. It regulates cardiac function through norepinephrine (NE) and neuropeptide Y and increases the heart rate (HR) through cAMP-mediated mechanisms in the sinoatrial node [35]. It also results in enhanced contractility and relaxation and increased atrioventricular conduction [36,37,38], and it alters ventricular repolarization (QT variability) [5,9,39,40] (Figure 2).

Notably, recent research suggests that a relationship exists between CAN activity and the left ventricular ejection fraction (LVEF). While the association between right insular activity and sympathetic activity is not new, it is thought that it could also be causally involved in cardiovascular disorders characterized by a reduced LVEF [32].

On the other hand, the parasympathetic pathway starts from the nucleus ambiguus (NAmb) and the dorsal motor nucleus of the vagus (DMNX) in the medulla and projects to postganglionic neurons that innervate the conduction system, the atria, and the ventricles [41,42]. Postganglionic fibers release acetylcholine and vasoactive intestinal peptide (VIP) [35,43] and exert their effects via M2 receptors that reduce cAMP and open potassium channels. Altogether, these changes hyperpolarize cardiac cells and decelerate cardiac function [5,44] (Figure 3). Parasympathetic tone is dominant at rest and is responsible for the lower resting HR compared with the intrinsic cardiac rate [5,9,45].

While the two branches of the ANS exhibit antagonistic effects on cardiac myocytes through cAMP signaling, parasympathetic tone can also inhibit the release of norepinephrine presynaptically [46]. These antagonistic interactions are linear when modulating for a mean HR but nonlinear in other phenomena like baroreflex sensitivity and HRV [47]. Respiratory sinus arrhythmia, for example, may involve both sympathetic withdrawal and sympathetic enhancement [5].

### 3.2. Neuroimmune Cardiovascular Interface (NICI)

Mediated by the immune system, another important point of interaction between the heart and the brain is the nerve–artery–immune interface (NICI), which consists of peripheral nerves, the adventitia of arteries and immune cells. Immune organs are directly innervated and thus linked to the nervous system. The thymus receives innervation via the sympathetic system, lymph nodes are subject to afferent signaling via dorsal root ganglia, the spleen is innervated by the splenic nerve, and the celiac ganglion and mucosa-associated lymphoid tissues (MALTs) are innervated by the sympathetic system while also utilizing neuropeptide signaling via VIP and substance P [41,48] (Figure 4). The NICI forms in healthy and diseased tissues but becomes particularly important in disease states, commonly triggered by the presence of an atherosclerotic plaque where it plays a role in promoting inflammatory responses, arteriogenesis via vascular endothelial growth factor (VEGF), and cardiac repair and remodeling post-ischemia [41,49].

### 3.3. Intrinsic Cardiac Nervous System (ICNS)

The intrinsic cardiac nervous system (ICNS), also called the “little brain of the heart”, is a peripheral network of ganglia and neurons integrated in the surface of the heart. These ganglia receive both sympathetic and parasympathetic input while also integrating signals from sensory afferent fibers and local interneurons [50]. The rostral ventriculolateral medulla (RVLM) provides sympathetic input, which increases NE release and drives HR, inotropy, systemic vascular resistance, and venous return [51,52,53]. Parasympathetic signals received from the rostral ventriculomedial medulla (RVMM), the hypothalamus, and the pons travel via the paravertebral ganglia and modulate HR and conduction [53,54]. The ICNS also receives parasympathetic input that initiates in the nucleus ambiguous and brainstem within the dorsal motor nucleus and travels through the vagus nerve [55].

### 3.4. Molecular Mechanisms

In pathologic states, systemic inflammation becomes a key mediator of heart–brain communication. It increases the levels of pro-inflammatory molecules like interleukin-1β (IL-1β), interleukin-6 (IL-6), and tumor necrosis factor (TNF-α), which drive endothelial dysfunction, neuroinflammation, cardiac fibrosis, and heart failure [9,15,16,17,18,19,56] and can affect important regulatory areas like the insular cortex, thus exacerbating cardiac dysregulation [16].

Extracellular vesicles packed with microRNAs (e.g., miR-124-3p), long noncoding RNAs (lncRNAs), and circular RNAs, which cross the blood–brain barrier (BBB), are also molecular mediators of inflammation, fibrosis, and even cardiac remodeling [19,57].

The gut barrier is another important parameter implicated in this axis. In pathologies like stroke, gut barrier integrity is disrupted, and bacterial endotoxins like lipopolysaccharides (LPSs) can enter the circulation and induce systemic inflammation. This worsens cardiac outcomes and highlights the implication of microbial and metabolic parameters in neurocardiac pathology [16,58].

Notably, shared molecular mechanisms have been identified between heart failure secondary to cardiac amyloidosis and Alzheimer’s disease. These include the upregulation of β-secretase-1 (BACE1), accumulation of β-amyloid peptide, and increased levels BACE1-AS, an amyloidogenic noncoding RNA [19,59].

### 3.5. Neurochemical and Hormonal Interactions

Regulation of cardiovascular function via reflex and feedback mechanisms incorporates not only neural but also neuroendocrine pathways.

The ANS directly modulates the cardiovascular system via the release of neurotransmitters like NE from the sympathetic and acetylcholine (ACh) from the parasympathetic that act on b1-adrenoreceptors and cholinergic receptors, respectively. Cardiomyocytes themselves can produce ACh and play a role in the regulation of cardiac activity [60,61]. Autonomic imbalance with sympathetic overactivity is a hallmark for pathologic states like HTN, HF, and arrhythmias [9].

In addition to NE release, the SNS also activates the renin–angiotensin system (RAS) and via increased levels of renin, angiotensin II (ATII), and aldosterone, induces vasoconstriction, inflammation, cellular perforation, and apoptosis. These effects along with mechanisms like neuroinflammation, oxidative stress, and RAS gene polymorphisms are important contributors to the pathogenesis of cardiovascular diseases as well as depression [4].

During stress, the hypothalamic–pituitary–adrenal (HPA) axis and sympathetic–adrenal–medullary (SAM) system also participate in feedback control. HPA axis activation leads to cortisol release and results in sodium retention, hypertension, and dyslipidemia while SAM activation leads to catecholamine release from the adrenal medulla. The HPA axis also plays a key role in neuroautoimmune diseases, like MS. In early stages, cortisol responses to stress are elevated while later, there is HPA axis blunting and reduced glucocorticoid secretion, contributing to disease progression [60]. These mechanisms directly link physiological stress to both cardiovascular and neurologic risk and clearly reflect the key presence and role of HBA in the real world of disease initiation [9] (Figure 5).

Overall, sympathetic activation and the resulting elevated levels of renin, angiotensin, cortisol, serotonin, and free radicals can result in serious clinical outcomes like cardiac arrhythmias, myocardial infarction (MI), sudden cardiac death (SCD), and QT prolongation [8]. Parasympathetic activation on the other hand leads to elevated levels of ACh, dopamine, endorphins, nitric oxide, and CoQ10, which all seem to have cardioprotective effects. It has also been shown that vagal stimulation can prevent stress-induced arrhythmias, particularly ventricular fibrillation, which further confirms the therapeutic potential of enhanced parasympathetic activity in cardiovascular diseases, with practical utility in existing and emerging wearables [8,62,63] (Figure 5).

Neuropeptide Y also plays a role in heart–brain communications. It is produced both in the CNS—especially in areas like the hypothalamus, amygdala and brainstem—and by neurons in peripheral tissues including the adrenal glands, sympathetic nervous system, and enteric nervous system in the GI tract [32].

### 3.6. Reflex and Feedback Mechanisms of the Heart–Brain Axis

Reflex pathways are a core component of the HBA, allowing for rapid cardiovascular adaptations. The baroreflex, mediated by mechanoreceptors expressing piezo proteins 1 and 2 (PIEZO1 and PIEZO2), is key in regulating blood pressure and maintaining autonomic balance. These proteins detect various types of mechanical stress, and these stimuli are converted to electrical signals that are communicated to the brain [32,64]. Carotid baroreceptors project to the nucleus tractus solitarius (NTS) and via glutaminergic output, they result in parasympathetic activation and sympathetic inhibition [41,65]. Dysregulation of this pathway is linked to diseases like HTN, tachycardia, orthostatic dysregulation, and HF [5,41].

Respiratory activity is also important in heart–brain communication. Information on all the main indices of the cardiorespiratory system is sent as afferent signals to the CAN, especially the NTS, insula, and locus coeruleus [60,66], via stretch receptors and baroreceptors [60,67].

The Bezold–Jarisch reflex, mediated by neuropeptide Y and cardiac NPY2R neurons, triggers rapid vagal amplification and potentially syncope in response to a lack of blood in the ventricles [32].

The chemoreflex utilizes chemoreceptors that detect changes in pO2, CO2, and pH levels and regulates both breathing and autonomic cardiovascular function [46]. Via projections to the NTS, it activates both the NAmb and the RVLM to simultaneously induce bradycardia and preserve perfusion [60,68].

The sensory circumventricular organs (CVOs) of the lamina terminalis—the subfornical organ (SFO), organum vasculosum lamina terminalis (OVLT), and median preoptic nucleus (MnPO) [69]—are highly vascularized areas that lack BBB control and restriction and participate in osmolality sensing, angiotensin II signaling via AT1 receptors, and vasopressin/oxytocin release. Notably, the MnPO receives signals from the OVLT, SFO, vagus nerve, and carotid baroreceptors and responds to hyponatremia and HTN by modulating sympathetic activity [5,6,8].

## 4. From Heart–Brain Axis Mechanisms to Therapeutic Strategies

Describing the mechanisms that comprise the HBA has laid the foundation for the development of therapeutic strategies while further understanding of their complex molecular pathways is paving the way for novel treatments, **some of which already implicate AI means, with integration and supervision.**

### 4.1. β-Blockade

Sympathetic hyperactivity, a hallmark of HBA dysregulation, underlying most axis-related disorders, especially neurogenic stunned myocardium (NSM), and takotsubo cardiomyopathy (TC) are already cornerstones of therapeutic strategies. β-blockers inhibiting b1 and b2 receptors reduce myocardial stress, oxygen demand, arrhythmogenic risk, and heart rate and suppress inflammatory cytokines like TNF-α [8,9,10,16,70], therefore improving survival outcomes [8,10,71].

### 4.2. Anti-Inflammatory Molecules

Inflammatory cytokines such as IL-1β, IL-6, and TNF-α can serve as therapeutic targets themselves. Cytokine blockers, NSAIDS, immune-modulating molecules, and multi-drug regimens have proven effective in cases like depression, Alzheimer’s, and heart failure by blunting the effect of those molecules, reducing endothelial and myocardial damage, and limiting neuroinflammation [9,15,16,17,18,19].

### 4.3. Vagus Nerve Stimulation (VNS) and Neuromodulation

Vagus nerve stimulation (VNS), invasive or transcutaneous, is FDA-approved for epilepsy and depression and under study for HF, postural orthostatic tachycardia syndrome, and stroke [18,72]. In epilepsy, it activates vagal afferents to the NTS, increasing NE and serotonin in the locus coeruleus and raphe magnus, disrupting seizure circuits and reducing IL-1β/TNF-α; the stellate ganglion blockade and epicardial injections further improve autonomic control in NSM and neurogenic arrhythmias [10,18,23,73,74]. In depression, benefits stem from monoaminergic enhancement, limbic–cortical modulation, lower pro-inflammatory cytokines, and higher BDNF [23,75]. In HF/HTN, VNS upregulates ACh, M3ACh receptors, and α7nACh receptors and suppresses cytokines, thus improving HR regulation, endothelial function, and infarct size [76,77]. In hypertensive rats, it prevents vascular stiffness and inflammation [23,73]. In stroke/TBI, it modulates ACh/cytokines via the cholinergic anti-inflammatory pathway and is associated with reductions in TNF-α and corrected ghrelin imbalances post-TBI [73,78]. Understanding heart–brain interactions and leveraging those mechanisms has paved the way for the development of wearables and AI models that aid in the diagnosis, monitoring, and treatment of their associated disorders.

### 4.4. Neurotrophin Modulation

Modulating neurotrophins—the brain-derived neurotrophic factor (BDNF), glial cell line-derived neurotrophic factor (GDNF), and ciliary neurotrophic factor (CNTF)—support neurogenesis, neuroprotection, and synaptic plasticity in neurologic disorders [24,75]. BDNF, being the most studied, enhances neuronal survival, differentiation, and long-term potentiation—key in Alzheimer’s disease (AD), Parkinson’s disease (PD), Huntington’s disease (HD), MS, traumatic brain injury (TBI), and spinal cord injury (SCI). In AD, it aids memory and synaptic repair. In HD, it preserves dopaminergic neurons while reduced BDNF and elevated pro-BDNF correlate with disease progression [24]. GDNF supports dopaminergic and motor neuron survival in PD and amyotrophic lateral sclerosis (ALS) [24,79], while CNTF may promote remyelination and motor/oligodendrocyte viability in ALS and SCI [24,80].

Direct BDNF therapy is limited by poor stability and BBB penetration. Instead, alternative strategies using TrkB agonists (7,8-DHF), which mimic BDNF signaling [75,81,82] and small-molecules like curcumin, resveratrol, and J-147 that boost BDNF and exert anti-inflammatory and antioxidant effects, are being explored [75].

### 4.5. Gut Barrier and the Heart–Brain Axis

The gut barrier and microbiota modulate systemic inflammation and neuroimmune balance [16,20,21,22]. Small chain fatty acids, especially butyrate, upregulate tight junctions, inhibit TLR4/NF-κB, lower circulating LPS and pro-inflammatory cytokines, strengthen barrier integrity, and regulate BBB permeability and microglial activation [20,21,22]. Neurotransmitters (GABA, serotonin, and catecholamines), vagal signaling, and the HPA axis are microbiota-dependent, making gut integrity crucial for heart–brain communication. Prebiotics, probiotics, and dietary fiber help restore homeostasis and mitigate neurologic and cardiac dysfunction [16,21,22].

### 4.6. RNA-Based Therapeutics

Circulating microRNAs, lncRNAs, and circRNAs regulate epigenetics, stress responses, and inflammation and can cross the BBB, acting as biomarkers and therapeutics in HBA disorders [19,83]. miR-124-4p rises after cardiac arrest, predicting survival and neurologic outcomes; BACE1-AS is shared by HF and AD, linking neurodegeneration and cardiac dysfunction [19,84,85]. Delivery via exosomes or other vesicles may suppress remodeling and promote repair in cardiac and neurologic tissues [23,86].

### 4.7. Nanoparticles

Nanoparticles (NPs) are applied in imaging, therapy, immune modulation, gene delivery, and regeneration across HBA-related disorders [87,88,89,90,91,92,93,94,95,96,97,98,99,100,101,102,103,104,105,106,107,108,109,110,111,112,113,114,115,116,117,118,119,120]. Superparamagnetic iron oxide nanoparticles and ultrasmall superparamagnetic iron oxide nanoparticles track inflammation and BBB integrity in MS and atherosclerosis, while gold NPs, silica NPs, and carbon nanotube-based sensors enable molecular and cardiac imaging [28,29,87,88,89,90,91,92,93,94,95,96]. Polymeric and lipid-based NPs improve the delivery of anti-hypertensives, anti-depressants, and neuroprotectants with enhanced CNS penetration, cardiac uptake, and immune regulation [87,88,89,90,91,92,93,94,95,96,97,98,99,100,101,102,103,104,105,106,107,108,109,110,111,112,113,114,115]. They also support gene/miRNA therapies [87,89,92,93,94,95,96,97,98,99,100,101,102,103,104,105,106,107,108,109,110,111,112,113,114,115,116,117,118] and regenerative strategies such as myelin repair, angiogenesis, and synaptic restoration [87,88,89,90,91,92,93,94,95,96,97,98,99,100,101,102,103,104,105,106,107,108,109,110,111,112,113,114,115,116,117,118]. Integrated with AI, NP-based biosensors and delivery systems offer real-time monitoring, prediction, and personalized treatment of neurocardiac disorders.

### 4.8. Targeting Brain Regions and Central Autonomic Circuits

Targeting the insular cortex, amygdala, and anterior cingulate cortex—key in sympathetic hyperactivity—may prevent cardiovascular consequences. Strategies include glucocorticoid receptor blockers, β-blockers, and anti-cytokine agents [5,9,17], as well as non-pharmacologic interventions like cognitive behavioral therapy, mindfulness, and anti-stress techniques, which reduce sympathetic tone and enhance parasympathetic activity [75,76,77,78,79,80,81,82,83,84,85,86,121,122,123,124].

## 5. Biomarkers of the Heart–Brain Axis and Their Utility in Clinical Practice

### 5.1. Heart Rate Variability (HRV)

Heart rate variability (HRV) is the key biomarker of the axis, reflecting autonomic nervous system balance, and it is influenced by both neurologic and cardiovascular diseases [5,9,10] (Table 1).

High resting HRV reflects autonomic flexibility and stress resilience, whereas low HRV indicates poor autonomic control and neurocardiac dysfunction [7]. Although not yet routine, HRV shows promise as a prognostic and diagnostic tool: low HRV predicts higher cardiac mortality and worse outcomes [5,9], while serial assessments may support early detection and monitoring in high-risk individuals [5,11,12,13]. In MS, impaired HRV is frequent, and it is associated with CAD, higher disability scores, disease activity, and inflammation [11,12]. Contradictory findings include associations of higher HRV with AF in hypertension [124], poorer behavioral regulation in autism [125], both high and low HRV with AF and CVD risk in general populations [126], and higher HRV with greater perceived life stress [127]. Overall, HRV interpretation appears population-dependent, suggesting a bimodal, rather than linear, risk pattern [124,125,126,127,128]. Here, AI comes to bridge the gap, confer extremely important tools, for diagnosis, follow-up, and even remote interventions, in a future upscale level, and indicate itself as a lifesaver.

### 5.2. Electrocardiographic (ECG) Changes

ECG abnormalities including QT prolongation, ST elevation/depression, T wave inversion, and ventricular or supraventricular arrhythmias are important indicators of neurocardiac dysfunction [10]. They are common in various disorders (Table 1) [14] and reflect disrupted autonomic regulation and myocardial instability from brain injury (e.g., stroke, and hemorrhage). Clinically, such patterns help distinguish neurogenic from primary cardiac dysfunction like MI [10], and markers such as QTc interval may have prognostic value for cardiovascular risk [12,13], where AI utility seems very important (Figure 6).

### 5.3. Cardiac Enzymes

Troponin: In brain injury, sympathetic overactivation can result in myocardial damage and elevated troponin even in the absence of coronary occlusion, making it a potential indicator of neurogenic cardiac injury [10]. While typically seen in NSM and TC, it has also been documented in stroke patients without MI [8] and in MS, where elevations along with ECG abnormalities during stress cardiomyopathy may raise suspicion of a relapse [14].

Creatine kinase (CK): Moderate CK elevations have been observed in ischemic stroke in the absence of MI, possibly due to minor myocardial injuries or noncardiac sources; however, its low specificity limits diagnostic use [8] (Figure 6).

### 5.4. Brain Natriuretic Peptide (BNP)

Elevated BNP and NT-proBNP have been documented in not only stroke, where they are associated with systolic dysfunction, AF, and larger infarcts, but also in cases of NSM and TC [8,10], where sympathetic overactivation leads to regional wall motion abnormalities, ventricular dysfunction, and ultimately, BNP secretion. BNP is thus a valuable biomarker with diagnostic and prognostic utility (Table 1), especially with the integration of AI models [8] (Figure 6).

### 5.5. Other Biomarkers

Several other biomarkers implicated in the HBA including BDNF, Myoglobin, IL-6, C-reactive protein (CRP), and TNF have been associated with pathological states, although their clinical utility remains limited [8,9,21] (Table 1).

**Table 1 ijms-26-08217-t001:** Summary of heart–brain axis biomarkers and their clinical utility. (HRV): heart rate variability; (MS): multiple sclerosis; (MI): myocardial infarction; (CNS): central nervous system; (ECG): electrocardiogram; (NSM): neurogenic stunned myocardium; (TC): takotsubo cardiomyopathy; (BNP): brain natriuretic peptide; (BDNF): brain-derived neurotrophic factor.

Biomarker	What It Reflects	Clinical Utility	Associated Conditions	References
HRV	Autonomic balance	Detection of autonomic dysfunction. Prediction of cardiac mortality and clinical deterioration.	Stroke, MS, depression, MI, CNS injury, diabetic neuropathy	[5,7,9,11,12,13,124,125,126,127,128]
ECG changes	Autonomic dysfunction, myocardial electrical instability	Differentiation between neurogenic and primary cardiac injury. Cardiovascular risk assessment.	NSM, TC, MS	[10,12,13,14]
Troponin	Neurogenic cardiac injury	Identification of neurogenic origins of cardiac injury.	NSM, TC, stroke, MS	[8,10,14]
Creatine Kinase	Neurogenic myocardial Injury	Limited diagnostic and prognostic utility.	Ischemic stroke	[8]
BNP	Autonomic and hemodynamic compromise	Identification of paroxysmal AF in stroke patients. Differentiation between neurogenic and primary cardiac injury. Prognostication of stroke recurrence, mortality and functional recovery.	Stroke, NSM, TC	[8,10]
BDNF	Susceptibility to neurocardiac stress	Limited clinical utility.	N/A	[9]
Myoglobin	N/A	Limited clinical utility.	Ischemic stroke	[8]
Inflammatory markers (CRP, IL-6, TNF)	N/A	Limited clinical utility.	Depression, neuroautoimmune and cardiovascular disorders	[9,21]

## 6. Autonomic Dysfunction as the Key Mechanism in Heart–Brain Axis Disorder

Autonomic dysfunction in the form of either failure or hyperactivity is the key mechanism implicating the heart–brain axis in a wide variety of disorders. Autonomic failure often accompanied by sympathetic impairment is a frequent feature of neurodegenerative disorders, especially synucleinopathies like Parkinson’s disease and multiple system atrophy [5,129]. Sympathetic hyperactivity, whether chronic or paroxysmal, can be the result of vascular, inflammatory, or traumatic injuries, adverse drug reactions, and neurologic or systemic conditions like obesity, HTN, diabetes, and anxiety [5]. This autonomic imbalance and particularly sympathetic overactivation is heavily implicated in the pathophysiology of various disease states. Disruption of reciprocal communication can have profound cardiovascular effects, including an increased risk for SCD and overall morbidity.

These observations highlight the significance of the HBA not merely as a communication network but as a dynamic system whose dysregulation is central to various cardiovascular and neurological disorders.

Key paradigms of these disorders are elaborated in the following sections.

### 6.1. Multiple Sclerosis and the Heart–Brain Axis

Autonomic dysfunction is an important feature of MS with up to two thirds of MS patients reporting cardiovascular autonomic dysfunction ranging from mild to severe grade [12]. This occurs via demyelination of the CNS and disruption of the CAN and is associated with lesions in the midbrain, parietal lobes, and the medulla oblongata [11,12,13,14]. It presents with decreased HRV, abnormal blood pressure responses during orthostatic challenges, and impaired systolic and diastolic pressure regulation during tilt-table testing [11,12].

In MS, both mechanisms of autonomic dysfunction seem to be present, each one at different phases. Sympathetic hyperactivity is more prominent during earlier stages and has been associated with supratentorial lesions like those in the insular cortex and hippocampus. Later stages, on the other hand, are characterized by noradrenergic failure and adrenergic receptor dysfunction, which also contribute to disease progression. It has been assumed that increasing impairment of the ANS during MS modulates immune responses, causing a vicious neuro-inflammatory cycle [13].

In rare cases, the disruption of these pathways can present with acute cardiac complications like TC and NSM. The culprit in both those syndromes is excessive sympathetic activation with documented cases reporting lesions in the medulla oblongata, the region where the cardiovascular center responsible for regulating HR, myocardial contractility, and vascular tone resides [7,14].

Notably, in a previous case report, a patient with MS presented with ST elevations and right bundle branch block (RBBB) on ECG, consistent with Brugada sign. This was the first time the coincidence of a MS patient with an asymptomatic BrS Type I in ECG was reported. These findings were attributed to ANS dysfunction and especially sympathetic failure due to demyelination of neurons at the inferior cervical spine and the superior thoracic level (T1–T2) [129].

The various anti-stress techniques, like meditation, mindfulness, muscle relaxation, art therapy, simple walking, etc., seem to be an everyday additional tool in the hands of neurologists, cardiologists, endocrinologists, and other professionals that are involved in health care promotion. In a single-case study, a MS patient practicing a mindfulness-based body scan technique nightly showed improvement in subjective sleep quality even though objective sleep biomarkers measured by Fitbit Sense 3 remained the same [130].

Another observational study found that the Pythagorean self-awareness intervention (PSAI)—a cognitive behavioral technique—lead to significant improvements in cognitive processing speed, verbal memory, stress, depression, anxiety and fatigue, further highlighting the potential of non-pharmacological interventions in managing a wide range of disorders [131].

### 6.2. Hypertension

Disruption of the HBA is also a key component of hypertension. HTN is closely regulated by the ANS, particularly the sympathetic, and according to microneurography studies, disease severity correlates with elevated sympathetic activity [41]. Sympathetic hyperactivity leads to elevated levels of catecholamines which in turn activate the RAS resulting in vasoconstriction and water retention [9]. ATII specifically, activates the subfornical organ (SFO) and OVLT and increases sympathetic tone and reactive oxygen species, thus contributing to elevated blood pressure. Mediated via chronic activation of the RVLM, increased sympathetic tone is also responsible for greater hypertensive responses during stress states, such as exercise and high salt intake [41,132]. Baroreceptor dysfunction, a key component of the HBA, whether afferent (vagal/glossopharyngeal damage) or efferent (diabetic neuropathy) can also lead to hypertension or orthostatic hypertension [41].

### 6.3. Atherosclerosis

The HBA is implicated in atherosclerosis through activation of the neuroimmune cardiovascular interface (NICI), where inflammatory signals from atherosclerotic plaques activate CNS sympathetic pathways, attenuate plaque progression, and stimulate bone marrow stem cells and inflammatory responses [8]. Stress-induced amygdala activity is associated with arterial inflammation, high-risk coronary plaques, and future risk of cardiovascular disease, linking emotional stress and atherosclerotic cardiovascular disease (ASCVD) [41,133]. Axonal sprouting near plaques along with nerve fiber infiltration by immune cells both amplify the disease while macrophage-derived netrin-1 promotes atherogenesis and is thought to be involved in axon guidance and arterial innervation [134,135]. It seems that new targeting but already existing or novel immunotherapies could soon eliminate this devastating disease, globally.

### 6.4. Stroke-Induced Cardiac Dysfunction

Stroke disrupts cardiovascular regulation via HBA mechanisms, mainly via sympathetic overactivation with insular involvement. Right insula lesions increase sympathetic tone and are associated with tachycardia and arrhythmias while left insular lesions increase parasympathetic activity leading to bradycardia and hypotension and reduce HRV, raising cardiac death risk [8,9,10]. The CAN, particularly areas like the insula, hypothalamus, and brainstem, are heavily involved in stroke-induced cardiac events including arrhythmias, MI, baroreflex failure, and SCD [5,8,10].

Common stroke-associated arrhythmias include AF, QT prolongation, ventricular tachycardia, sinus node dysfunction, and ventricular fibrillation [8]. Stroke may also lead to diastolic dysfunction and increased left ventricular end-diastolic pressure, resulting in increased sympathetic signaling, endothelial injury, and hypercoagulability [9].

### 6.5. Heart Failure

In chronic heart failure, sympathetic overactivity activates the RAS, elevates catecholamine, increases myocardial oxygen demand, and drives ventricular remodeling [4]. In heart failure with reduced ejection fraction (HFrEF), decreased intracranial perfusion (particularly in subcortical areas) is associated with arrhythmias, implantable cardioverter–defibrillator (ICD) therapy administration, and cardiac arrests [136]. This risk is attributed to autonomic imbalance and QTc prolongation [136], which clearly indicates the involvement of HBA mechanisms.

### 6.6. Takotsubo Cardiomyopathy

Takotsubo cardiomyopathy is an example of the prominent effects of sympathetic hyperactivity, mediated by heart–brain communication. Emotional or physical stress causes an acute catecholamine surge that presents as a condition that mimics MI and involves apical ballooning, arrhythmias, and in severe cases, cardiogenic shock and SCD. The brain regions underlying this syndrome are thought to be the limbic system (insula, hippocampal gyrus, amygdala, etc.), ventral medial prefrontal cortex, and brainstem [9,137].

### 6.7. Depression

Depression, marked by chronic stress, highlights HBA communication via its association with cardiovascular disease [9]. In this case, autonomic imbalance results in decreased HRV and activation of the HPA axis, SAM system, RAS, and inflammatory pathways, thus elevating levels of cortisol, ATII, aldosterone, IL-6, TNF, and CRP. In MS, mental disorders including depression are frequent and strongly associated with autonomic imbalance [60]. These changes exacerbate coronary heart disease, chronic HF, arrhythmias, and overall cardiovascular (CVD) risk. Notably, HRV monitoring in depressed patients can be used as biomarker to assess cardiac risk [9].

Transcranial alternating current stimulation (tACS) delivered by a wearable home-used device shows promise for major depressive disorder, improving Beck depression inventory-II (BDI-II) scores with high adherence and minimal adverse effects in clinical trials [138].

### 6.8. Other Disorders

Several other conditions exhibit features that highlight the complex role of dynamic heart–brain interactions including hippocampal pathology [31], obstructive sleep apnea syndrome and other sleep disorders [139,140], Guillain–Barré [141,142], baroreflex failure syndrome [5], fatal familial insomnia [143,144], and sepsis [145,146,147,148,149].

## 7. Heart–Brain Axis in the Age of Wearables and AI

The rise in wearables and AI systems has profoundly changed the landscape and allowed for innovative diagnostic, prognostic, and monitoring applications. Heart–brain axis mechanisms, through their associated biomarkers, can now be effectively leveraged and integrated into clinical practice, significantly improving patients’ quality of life.

### 7.1. AI in Multiple Sclerosis

In MS, deep learning (DL) and radiomic approaches applied to MRI enable automated lesion detection and segmentation, distinguish MS from mimicking conditions, and improve classification when combined with clinical data [138,139,140,141,142,143,144,145,146,147,148,149,150,151,152,153,154,155,156,157,158,159]. AI models predict conversion from clinically isolated syndrome to MS, progression to secondary progressive, and long-term disability using imaging and clinical features [148,149,150,151,152,153,154,155,156,157,158,159,160,161,162,163,164,165,166,167,168,169]. In monitoring, AI quantifies lesion load, brain atrophy, and therapy response across serial MRIs, while digital twins and wearable sensors extend tracking to real-time disease evolution and mobility [153,158,159]. Regarding therapy, AI predicts patient response to disease-modifying treatments, guides personalized treatment planning, accelerates drug discovery, and supports rehabilitation technologies such as robot-assisted gait training [153,158,159].

### 7.2. AI in Epilepsy

Across the epilepsy spectrum, machine learning (ML) and DL enhance EEG analysis for automated detection of seizures and interictal epileptiform discharges, while imaging-based AI identifies subtle lesions such as hippocampal sclerosis and cortical malformations, often surpassing expert review. Multimodal integration further improves localization of epileptogenic zones [160,161,162,163,164,165,166,167,168,169,170,171,172,173].

AI models can also predict seizure recurrence, drug resistance, and surgical outcomes, with imaging biomarkers such as hippocampal volumetry adding prognostic value [150,151,152,153,154,155,156,157,158,159,160,161,162,163,164,165,166,167,168,169,170,171,172,173]. They enable real-time EEG analysis and wearable seizure detection using multimodal biosignals, supporting outpatient and home-based care [150,151,152,153,154,155,156,157,158,159,160,161,162,163,164,165,166,167,168,169,170,171,172,173]. In regard to therapy, AI aids surgical planning and outcome prediction, powers closed-loop neuromodulation systems, and forecasts individual responses to anti-seizure medications, advancing personalized treatment [150,151,152,153,154,155,156,157,158,159,160,161,162,163,164,165,166,167,168,169,170,171,172,173].

### 7.3. HRV

#### 7.3.1. HRV—Wearables

ECG or photoplethysmography (PPG)-based HRV monitoring can aid the diagnosis of post-traumatic stress disorder (PTSD), anxiety, depression, HTN, HF, arrhythmias, and hypoglycemia [174] via detecting ANS dysfunction, a hallmark of HBA dysregulation. Devices like Oura, Fitbit, and HRV4Training can track stress and recovery [174,175,176] while some offer biofeedback interventions like breathing guidance and mindfulness techniques to regulate stress [176]. Devices like the Apple Watch and Microsoft Band 2 provide real-time stress monitoring used as a risk factor for accidents and cognitive impairment [177,178,179,180] (Table 2 and Figure 7).

Systems utilizing photonic, electrical, and optical sensing “modalities” (SMF photonic system [181], Wavelet wristband [182], and POF smart wristband [183]) can aid in detecting arrhythmias, vascular abnormalities, sleep apnea, pneumonia, chronic obstructive pulmonary disease (COPD), and more with high accuracy.

Beyond clinical applications, the WSB-900 combines ECG and optical sensing for multimodal biometric authentication and can be used for identity verification [184].

#### 7.3.2. HRV-AI

Machine learning and deep learning models can analyze HRV and extract patterns directly dependent on autonomic regulation that reflect heart–brain axis integrity. These can accurately detect various HBA-related disorders like AF, HF [185,186,187,188], depression, stress, cognitive load, and PTSD [185,186]. They can be used to predict the risk for seizures and neurocardiogenic injury, classify sleep stages, and detect sleep apnea [185,186,187,188]. Other models can assess the risk for adverse outcomes including sepsis, SCD, and cognitive decline [185]. Notably, cloud-based AI platforms like AI Watson have reported predictive accuracy levels of up to 90% for HF and other conditions [185].

### 7.4. ECG—AI

AI ECG analysis detects AF, AV block, VT, silent AF [189,190,191,192,193,194], LV dysfunction, hypertrophic cardiomyopathy, valvular disease, amyloidosis (AUC 0.95–0.98), conduction, and ischemic abnormalities with near 100% sensitivity [192,195].

Leveraging HBA mechanisms, cardiovascular signals from ECGs can also serve as proxies for neurologic dysfunction, as shown by AI models detecting idiopathic Parkinson’s disease from cardiac patterns.

Beyond the HBA, they can detect systemic conditions like hyperthyroidism, anemia, cirrhosis, electrolyte disorders, and COVID-19 [191,195,196]. Prognostically, they predict HF up to 10 years in advance and AF within 6 months to 10 years and perform risk stratification for SCD, stroke, and mortality, even in patients with normal ECGs [191,193,194,196,197].

### 7.5. Troponin—Wearables

Troponin, the golden standard for acute MI, can now be detected via wearables like the Infrasensor and the wrist-worn infrared spectrophotometric sensor, allowing for rapid, non-invasive MI detection and CVD risk assessment. This offers a fast and non-invasive alternative for monitoring cardiac troponin that eliminates the need for blood drawing and allows at-home or remote cardiac testing [198,199]. Given troponin’s role in identifying HBA-related disorders like neurogenic cardiac injury, NSM, TC, and stress cardiomyopathy in MS relapses, AI-enabled tools could facilitate the early detection, monitoring, and prognostication of these conditions [8,10,14].

### 7.6. Creatine Kinase—Wearables and AI

Wearables and AI models measuring CK offer valuable insights into HBA-related disorders as CK fluctuations reflect neurogenic myocardial injury triggered by acute brain insults or autonomic imbalance [8]. Devices like the Pcrea/Cys/Au-SPE sensor enable rapid MI diagnosis [200]; sweat sensors like the TPE-Gr can track CK in athletes and workers to assess muscle damage, heart-related stress, and cardiac events and even assist in personalized recovery tracking [201,202,203]. In certain cases, AI models were able to predict CK from AST/ALT levels [204] and in statin-treated patients, identify those at high risk of statin-induced muscle injury [125].

### 7.7. BNP-AI

In the context of HF, some noteworthy AI applications include a model using BNP that can predict cardiac events [205], another that estimates BNP levels from chest X-rays and allows early detection and improved diagnostic accuracy [206], and a model that predicts BNP and pro-BNP from ECG signals, enabling fast, low-cost screening [207]. Since BNP release is modulated by both cardiac wall stress and central neurohormonal signaling, these tools can serve as key indicators of HBA dysfunction [8,10].

## 8. Challenges and Limitations of AI and Wearable Technologies

While these technologies show significant promise, they still present several limitations and concerns. When it comes to models and wearables that process sensitive biometric data such as HRV and ECG, privacy and security risks become especially pronounced. Despite encryption and protective measures, such data is often transferred over nonsecure channels like emails or networks and cloud environments that lack robust data governance, leaving them vulnerable to security breaches [3,4,5,6,8,9]. Legal and ethical considerations remain unresolved and there are still no standardized frameworks regarding data transmission, population screening or the use of learning systems, resulting in uncertainty around accountability issues and regulatory compliance [4,5,6,8,9].

AI models, especially those using deep learning techniques, lack transparency and interpretability and are often termed as “black boxes”. Clinicians cannot understand how AI reaches a decision, thus limiting trust and interpretability [1,4,5,7]. AI ECGs may also rely on unknown or nonstandard features that would normally not be interpreted, which further aggravates transparency issues and poses the risk of overdiagnosis [1,5,6,7]. Additionally, most AI systems are trained using controlled, retrospective, or demographically narrow datasets and thus perform poorly when applied in real-world environments or diverse populations [7,10,11].

The integration of wearables further complicates these concerns. Devices like the Empatica E4 or Samsung Gear S2 suffer from motion artifacts, signal loss, and limited battery life, which impair their ability to continuously and reliably monitor key HBA biomarkers like the HRV and render them problematic in nonstationary, real-life conditions [12,13,14,15].

In regard to stress, its subjective and context-dependent nature makes wearable-derived metrics highly variable and limits models’ reliability [16,17,18]. The variability of devices themselves is also a challenge that needs to be addressed. Different brands of smartwatches or other wearables use proprietary, nonstandard algorithms, which result in inconsistent and variable results across platforms [17]. Finally, biological and clinical parameters that vary among individuals like age, race, gender, health status, and comorbidities affect both biomarker levels (e.g., troponin and BNP) and physiological responses and require stratification to mitigate bias, which is rarely implemented in current models [2,4,5].

In general, both algorithms and wearable devices urgently need a well-tested and accepted international regulatory framework, given the rapid development and spread of their use.

## 9. Discussion

The heart–brain (HBA) or brain–heart axis has been mentioned through philosophical and empirical approaches since ancient civilizations up to date. The immense and exponentially increasing knowledge at the molecular, electrophysiological, neuroanatomical, neuroimaging, neuroimmunological, and nowadays, at the artificial intelligence level has given us the opportunity to expand our ability to both diagnostic and therapeutic frame, since many neurological and cardiological disorders have their origin in HBA and autonomic dysfunction of various reasons.

Most important, the HBA and its components paved the way for wearables, since HRV or autonomic dysfunction were the first targets of wearables that emerged and still remain. Moreover, anti-stress techniques target the HBA in health and disease, and new technological means already give us the ability to have a brand-new intervention based on immune modulation and inflammation control, gene and miRNA delivery, nanoparticles, and regeneration via expanding AI utility.

Artificial intelligence (AI) gives us abilities for better diagnosis, follow-up, prognostication, and therapeutic intervention, much earlier than in the past, increasing the quality of life of patients and their families. Additionally, it adds to our curiosity of the HBA mysteries, now being revealed thanks to our imagination and also AI.

## Figures and Tables

**Figure 1 ijms-26-08217-f001:**
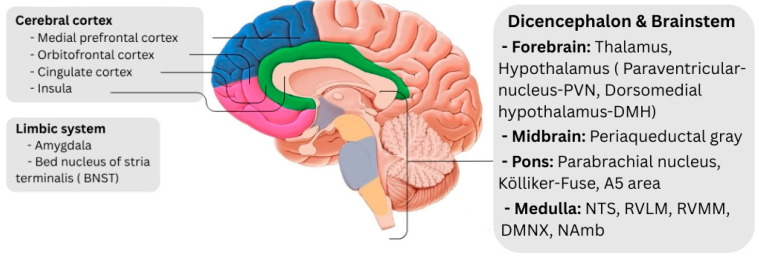
Main areas of the central autonomic network; (BNST): bed nucleus of stria terminalis; (PVN): paraventricular nucleus; (DMH): Dorsomedial hypothalamus; (NTS): nucleus tractus solitarius; (RVLM): rostral ventrolateral medulla; (RVMM): rostral ventromedial medulla; (NAmb): nucleus ambiguus; (DMNX): dorsal motor nucleus of the vagus.

**Figure 2 ijms-26-08217-f002:**
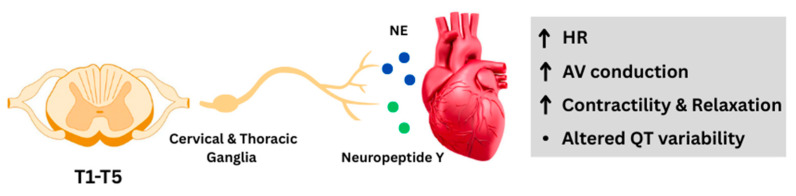
The sympathetic pathway in the heart–brain axis. (HR): heart rate; (NE): norepinephrine; (AV): atrioventricular; (↑): increase.

**Figure 3 ijms-26-08217-f003:**
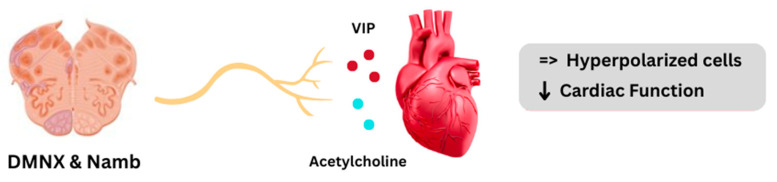
The parasympathetic pathway in the heart–brain axis. (DMNX): dorsal motor nucleus of the vagus; (NAmb): nucleus ambiguus; (VIP): vasoactive intestinal peptide; (↓): decrease.

**Figure 4 ijms-26-08217-f004:**
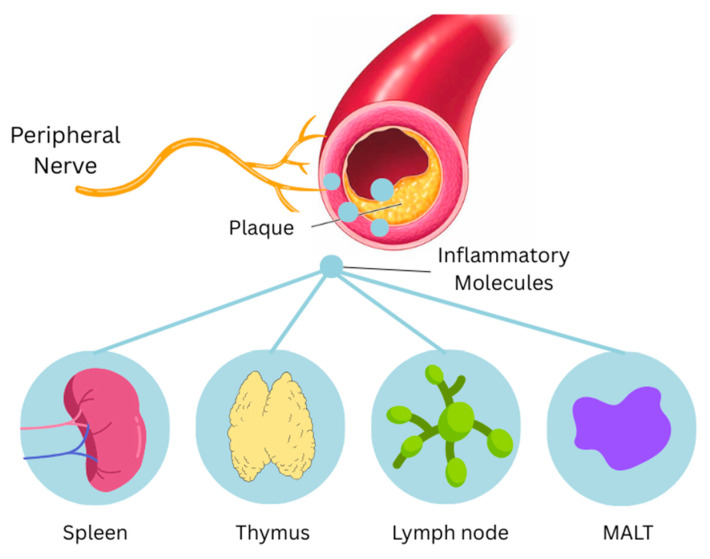
Activation of the NICI in the presence of atherosclerosis. (NICI): neuroimmune cardiovascular interface; (MALT): mucosa-associated lymphoid tissue.

**Figure 5 ijms-26-08217-f005:**
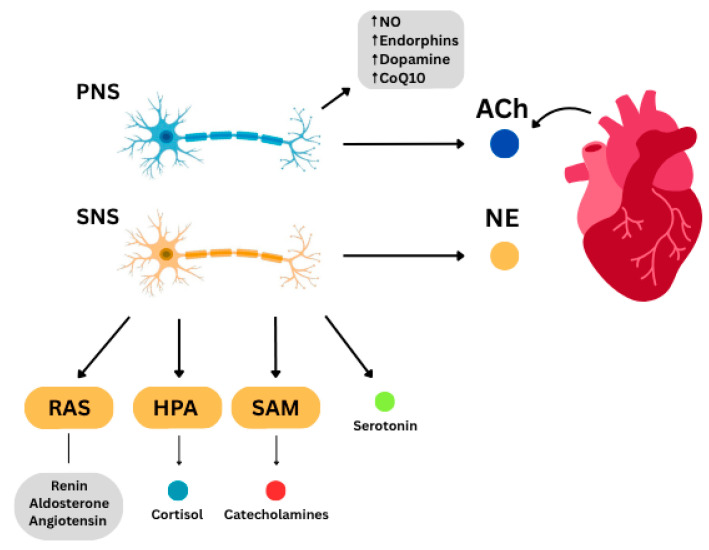
Neurohormonal and neurochemical pathways of the heart–brain axis induced by sympathetic and parasympathetic activation; (PNS): parasympathetic nervous system; (SNS): sympathetic nervous system; (RAS): renin–angiotensin system; (HPA): hypothalamic–pituitary–adrenal; (SAM) sympathetic–adrenal–medullary; (NO): nitric oxide; (CoQ10): coenzyme Q10; (ACh): acetylcholine; (NE): norepinephrine.

**Figure 6 ijms-26-08217-f006:**
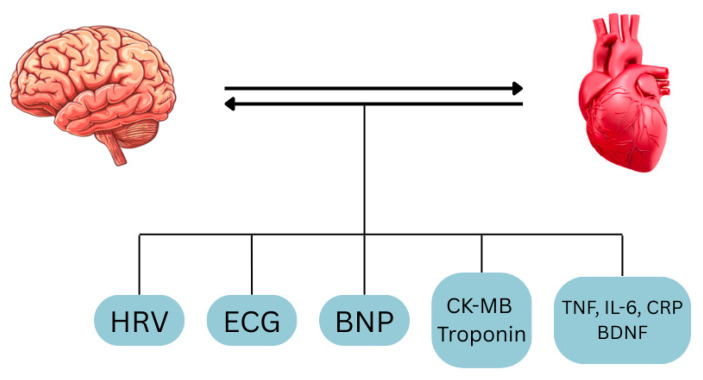
Main biomarkers of the heart–brain axis. (HRV): heart rate variability; (ECG): electrocardiogram; (BNP): brain natriuretic peptide; (CK-MB): creatine kinase; (TNF): tumor necrosis factor; (IL): interleukin; (CRP): C-reactive protein; (BDNF): brain-derived neurotrophic factor.

**Figure 7 ijms-26-08217-f007:**
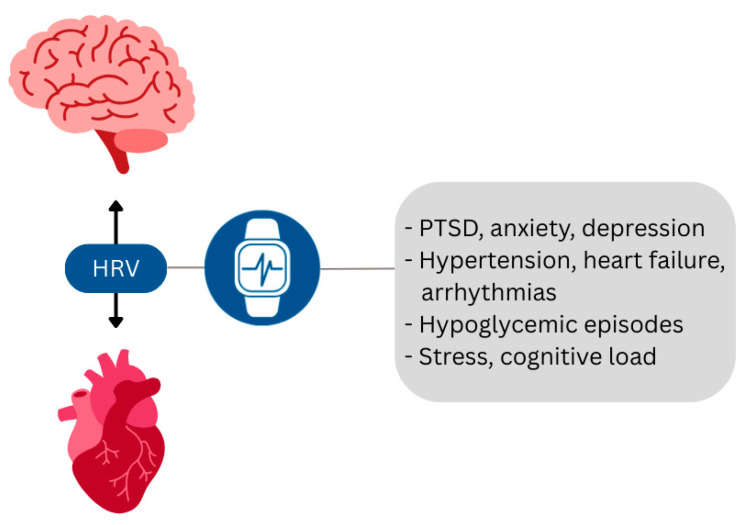
Biofeedback and diagnostic applications of wearables utilizing HRV monitoring. (HRV): heart rate variability; (PTSD): post-traumatic stress disorder.

**Table 2 ijms-26-08217-t002:** Summary of wearables for HRV monitoring. (ECG): electrocardiogram; (PPG): photoplethysmography.

Category	Devices/Apps
ECG-based Wearables	Welch Allyn Cardio Perfect Pro, Firstbeat Bodyguard 2, Bittium Faros 360, Actiheart, HexoskinProShirt, Aidlab, Equivital EQ-02, BITalino (r)evolution Kit, AIO Smart Sleeve, Polar H10, Polar H7, Zephyr Bioharness
PPG-based Wearables	Empatica E4, Samsung Gear S2, Microsoft Band 2, Apple Watch, Fitbit, Oura Ring, Whoop Band, Garmin Watches, Samsung Galaxy Watch, Polar Watches, Biovotion Everion, Polar OH1
Hybrid Apps	Elite HRV, Welltory, HRV4Training

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
