# Peer review of "The Heart–Brain Axis in the Artificial Intelligence Era: Integrating Old and New Insights Towards New Targeting and Innovative Neuro- and Cardio-Therapeutics"

_ijms, 2025, doi:10.3390/ijms26178217_

Round 1
Reviewer 1 Report
Comments and Suggestions for Authors
This review comprehensively explores the Heart-Brain Axis (HBA), highlighting its bidirectional communication via neural, hormonal, and immune pathways. The authors discuss the role of autonomic dysfunction in cardiovascular and neurological disorders, emphasizing biomarkers like HRV and ECG. They also review emerging therapies (e.g., vagus nerve stimulation, nanoparticles) and the transformative potential of AI and wearables in diagnostics and treatment. The paper integrates historical perspectives with cutting-edge research, offering a holistic view of HBA mechanisms and clinical applications. While thorough, the review could benefit from clearer structural organization and deeper critical analysis of conflicting evidence in the field. Several suggestions are supplied:
1. Suggest add a table summarizing key biomarkers and their clinical utility, if possbile
2. Suggest enhance discuss limitations of wearable/AI technologies (e.g., data privacy).
3. Suggest enhance the recent (2024–2025) references about Heart and Brain applications, such as Wearable photonic smart wristband for cardiorespiratory function assessment and biometric identification.
4. If poosibe, suggest Streamline the "History" section to focus on pivotal discoveries.
5. Suggest enhance address contradictory findings on HRV prognostic value.
Author Response
Dear reviewer,
We would like to thank you for carefully reviewing our manuscript and for your insightful comments and suggestions. Your feedback was highly valuable in improving the clarity, depth, and completeness of our work. We have revised the manuscript accordingly and incorporated all your recommendations. You will find the relative text highlighted with yellow colour for your convenience in the revised manuscript document.
Specifically:
- We streamlined the History section to focus on pivotal discoveries
- In the biomarkers section, we enriched the HRV subsection to include studies that report contradictory findings regarding HRV
- Added a table summarizing key HBA biomarkers and their clinical utility
- In the Wearables and AI section, we enriched the HRV wearables subsection to include photonic-based systems, including the wristband, for cardiorespiratory monitoring and biometric identification.
- Added the “ Challenges and Limitations of AI and Wearable Technologies” section where we analyze the current limitations these systems pose such as data privacy and security risks, absence of an international, well-established regulatory framework, the “black box” issue regarding AI applications, interpretability issues etc. \
Your comments were very helpful in crafting a more thorough and complete version of our review. We hope that you find our revised manuscript improved and responsive to your suggestions
Thank you for your time and input! We remain at your disposal for any further clarifications.

Reviewer 2 Report
Comments and Suggestions for Authors
-The manuscript purports to be a review, but the current organization of the content is confusing and unsystematic. The sections appear to overlap, and the literature analysis is often descriptive, not critical. A clear conceptual taxonomy (e.g., division into molecular, neural, immunological mechanisms, etc.) and a visual or tabular mapping of the cited studies are lacking. In several sections, the reader has difficulty understanding which points are consolidated, which are emerging, and which are still controversial. A thorough restructuring is recommended to foster narrative coherence and methodological rigor.
-Superficial treatment of Artificial Intelligence lacking concrete contributions
Despite the title promising an integration of AI into the context of HBA, the section dedicated to Artificial Intelligence is very weak, generic, and disconnected from the rest of the discussion. "Wearables" and "predictive models" are mentioned abstractly, without concrete examples, validated studies, architectures used, or methodological limitations. References to peer-reviewed publications documenting the actual use of AI in the context of HBA or HRV-based monitoring are lacking. At this stage, the inclusion of AI appears more like an attractive label than substantive content. A radical strengthening of this section or a rewording of the title is needed.
-A review should offer, in addition to a summary of the data, concrete insights into the future. This manuscript completely lacks a clear thematic conclusion, a critical reflection on the outstanding challenges (e.g., HRV signal heterogeneity, AI model validation, ethical limitations in the use of wearables), and a proposed future roadmap for preclinical research or integrated clinical applications.
The current conclusion is weak and leaves no scientifically relevant message.
-The title heralds an integration of classical pathophysiology and AI-supported therapeutic innovations, but the actual content falls short of this ambition. The "old" sections (e.g., HRV, autonomic axis, beta-blockers) are presented without innovative contributions, while the "new" sections (e.g., neurotrophins, vagus nerve stimulation, nanotherapies, AI predictive models) are treated superficially or with a speculative tone. The result is a review that appears uneven, poorly focused, and scientifically lacking in impact compared to the standards expected from a journal like IJMS.
Author Response
Dear reviewer,
Thank you for reading our manuscript, although you responded in a very strict tone and we certainly cannot agree on the general comments, but we have utilized all of your paragraphs to improve our manuscript, in accordance with the suggestions of the first reviewer, as well.
Our general replies are:
- It is not our first manuscript in the reputable journal ‘International Journal of the Molecular Sciences’, which we fully respect, that is why we submit this review which we think will make a severe contribution and offer to the journal and the readers
-The title of our manuscript is “The Heart-Brain Axis in the Artificial Intelligence Era: Integrating old and new insights towards new-targeting and innovative neuro- and cardio-therapeutics” and aims to integrating old and new insights, making references to AI means as well, but it is not a manuscript on AI per se ! That is why we start from historical points, which paragraph we have though shortened now in our revised manuscript, according to 1st reviewers suggestion.
-Our methodology is clear (paragraphs and sub-paragraphs, in numbers) from historical, to anatomical, to molecular, to neurophysiological, to neuroimmunological, to therapeutical frame, including new insights from gut-brain axis, nanothechnology etc in order to reach AI frame
-We clearly give a full table of wearables, which now we have enriched and extended, according to 1st reviewers suggestion.
-We give our experience and critical point of view, on Heart-Brain-Axis, and ANS, including our references on the articles we have already published. The last one in on wearables and HRV in Multiple Sclerosis, the autoimmune disease, we are involved, for many years
1. Gialafos E, Andreadou E, Kokotis P, Evangelopoulos ME, Haina V, Koutsis G, Kilindireas C, Filippatos G, Anagnostouli M. Brugada sign in a multiple sclerosis patient: Relation to autonomic nervous system dysfunction and therapeutic dilemmas. Int J Cardiol. 2016 Jan 1;202:652-3. doi: 10.1016/j.ijcard.2015.09.094. Epub 2015 Sep 26. PMID: 26451794.
2. Anagnostouli M, Markoglou N, Chrousos G. Psycho-neuro-endocrino-immunologic issues in multiple sclerosis: a critical review of clinical and therapeutic implications. Hormones (Athens). 2020 Dec;19(4):485-496. doi: 10.1007/s42000-020-00197-8. Epub 2020 Jun 1. PMID: 32488815.
3. Anagnostouli, M., Babili, I., Chrousos, G., Artemiadis, A., & Darviri, C. (2018). A novel cognitive behavioral stress management method for multiple sclerosis: A brief report of an observational study. Neurological Research, 41(3), 223–226. https://doi.org/10.1080/01616412.2018.1548745
4. Iliakis, I., Anagnostouli, M., & Chrousos, G. (2024). Assessing the impact of the mindfulness-based body scan technique on sleep quality in multiple sclerosis using objective and subjective assessment tools: Single-case study. JMIR Formative Research, 8, e55408. https://doi.org/10.2196/55408
Of course we accept that the limitations for AI utility was missed and we have added it now, the table on wearables has also enriched and extended
Generally according to the 1st reviewer suggestion we have:
1. Added a table summarizing key biomarkers and their clinical utility
2. Included limitations of wearable/AI technologies (e.g., data privacy).
3. Enhanced the recent (2024–2025) references about Heart and Brain applications, such as Wearable photonic smart wristband for cardiorespiratory function assessment and biometric identification.
4. Streamlined the "History" section to focus on pivotal discoveries.
5. Addressed contradictory findings on HRV prognostic value.
Round 2
Reviewer 2 Report
Comments and Suggestions for Authors
I appreciate the substantial effort made to address the initial review, particularly the inclusion of a biomarker table, the expansion of wearable/AI limitations, the update with recent (2024–2025) references, and the streamlining of the historical section. These additions improve the manuscript’s breadth and readability. However, some concerns remain:
-The manuscript still positions AI more as an “add-on” rather than an integrated component of the Heart–Brain axis discussion. A clearer alignment between the AI elements and the core neuro–cardio mechanisms would improve thematic coherence.
-The work remains closer to a narrative review than a systematic one. This should be clearly stated in the abstract and methods to set appropriate expectations for readers.
-While the manuscript presents valuable insights, several statements (particularly regarding mechanistic links and therapeutic implications) would still benefit from additional supporting references, where available, to strengthen credibility.
-The expansion in content has increased the amount of information but at times risks dispersion. A tighter synthesis of key points, especially in the therapeutic and AI-related sections, would enhance impact.
Overall, the revisions are constructive, but further refinement in framing, clarity of review type, and evidence support would substantially improve the scientific value and focus of the manuscript.
Author Response
Dear reviewer,
We would like to thank you for carefully reviewing our manuscript once again and for providing us with insightful comments and suggestions. Your feedback was very helpful in making some key revisions in our manuscript which we believe improved the clarity and structure of our work and overall made it more comprehensible and impactful. In our revised manuscript, we incorporated all your suggestions which you will find highlighted with yellow colour for your convenience.
Overall, we have shortened the length of the manuscript and made it more concrete, regarding the connection of each subparagraph with our final goal, AI, as it was your main suggestion.
Specifically, we:
- Made an effort to better align the AI components of our paper into the HBA discussion by:
- Adding subsections 6.1, 6.2 regarding AI applications in Multiple Sclerosis and Epilepsy, 2 disorders that are representative examples of autonomic dysfunction and thus HBA dysregulation,
- Including a brief sentence in subsection 6.3 - 6.7, briefly explaining how the HBA axis is linked to the AI/wearable application discussed each time ( you will find the sentences in bold, yellow text )
- Including a brief sentence in subsections 2.5, 4.1 -4.4 briefly explaining how AI fits into the HBA component discussed each time ( you will find the sentences in bold, yellow text )
- Highlighting the link between HBA components and AI/wearable techonology in our Discussion.
2) Stated, in the abstract, that the present review is a narrative one.
3) Enriched our references throughout sections 3-4 regarding HBA mechanisms and therapeutic strategies for further credibility ( you will find new references in bold, yellow text )
4) Made an effort to improve comprehensibility and clarity by:
- Condensing the information presented in sections 3, 4 & 6 regarding therapeutic strategies, biomarkers and AI/wearable applications along with subsections 2.6 & 5.
- Adding a reference column in Table 1 regarding biomarkers and their clinical utility so that readers can easily find more details if interested.
- Replacing section 8 -which included a detailed description of nanoparticles, their role and utilities-, with subsection 3.7, a condensed paragraph that states the key points regarding the clinical utility of nanoparticles in the context of the HBA.
- Rephrased a sentence in subsection 2.2 and figure 4 regarding the NICI, for clarity purposes.
Your comments were valuable in crafting a more thorough and complete version of our manuscript. We hope that you find our revised manuscript improved and responsive to your suggestions.
Thank you for your time and input. We remain at your disposal for any further clarifications.
Round 3
Reviewer 2 Report
Comments and Suggestions for Authors
I am pleased with the input from the review round. The work is now more substantial and deserving of publishing.